# Layer Swapping for Zero-Shot Cross-Lingual Transfer in Large Language Models

**Lucas Bandarkar**[§][*]          **Benjamin Muller**          **Pritish Yuvraj**
**Rui Hou**          **Nayan Singhal**          **Hongjiang Lv**          **Bing Liu**
Meta GenAI                    [§]University of California, Los Angeles

## Abstract

Model merging, such as model souping, is the practice of combining different models with the same architecture together without further training. In this work, we present a model merging methodology that addresses the difficulty of fine-tuning Large Language Models (LLMs) for target tasks in non-English languages, where task-specific data is often unavailable. We focus on mathematical reasoning and without in-language math data, facilitate cross-lingual transfer by composing language and math capabilities. Starting from the same pretrained model, we fine-tune separate "experts" on math instruction data in English and on generic instruction data in the target language. We then replace the top and bottom transformer layers of the math expert directly with layers from the language expert, which consequently enhances math performance in the target language. The resulting merged models outperform the individual experts and other merging methods on the math benchmark, MGSM, by 10% across four major languages where math instruction data is scarce. In addition, this *layer swapping* is simple, inexpensive, and intuitive, as it is based on an interpretative analysis of the most important parameter changes during the fine-tuning of each expert. The ability to successfully re-compose LLMs for cross-lingual transfer in this manner opens up future possibilities to combine model expertise, create modular solutions, and transfer reasoning capabilities across languages *all post hoc*.

## 1 Introduction

Instruction fine-tuning Large Language Models (LLMs) is necessary to customize pre-trained models for real-world applications. This fine-tuning is especially critical in multilingual settings because most popular open-source LLMs have been pretrained on highly English-centric data (Jiang et al., 2023; Llama et al., 2024; Yang et al., 2024a). Although recent LLMs such as Llama 3 and Qwen2 have seen more than a trillion non-English tokens due to the sheer scale of their pretraining datasets, these tokens are still heavily concentrated in just a few languages, resulting in limited capabilities for most other non-English languages (Llama et al., 2024; Yang et al., 2024a). Furthermore, the scarcity of high-quality labeled data available for post-training in non-English languages and the tremendous cost of procuring it further exacerbates the inequality. Even machine-translating English post-training data into target languages—a typical solution—has significant computational overhead and often leads to datasets of unreliable quality (Khanuja et al., 2024). Numerous efforts have set out to annotate or assemble high-quality datasets for lower-resource languages (Khan et al., 2024; Singh et al., 2024b; Tonja et al., 2024), but a massive gap still exists in many tasks and domains, such as math. As a result, developers are forced to largely rely on cross-lingual transfer—the generalization of learned capacities from high-resource languages to lower ones—but the rate of such transfer is low for most languages (Philippy et al., 2023).

In this paper, we present a novel solution that merges two LLMs together in order to transfer math reasoning capabilities to lower-resource languages during supervised fine-tuning (SFT). In the absence of in-language math data, we fine-tune two variants of the same pretrained model: one with English math samples and the other with generic instruction data in the target language. Provided

---

[*]Correspondence to `lucasbandarkar@cs.ucla.edu`

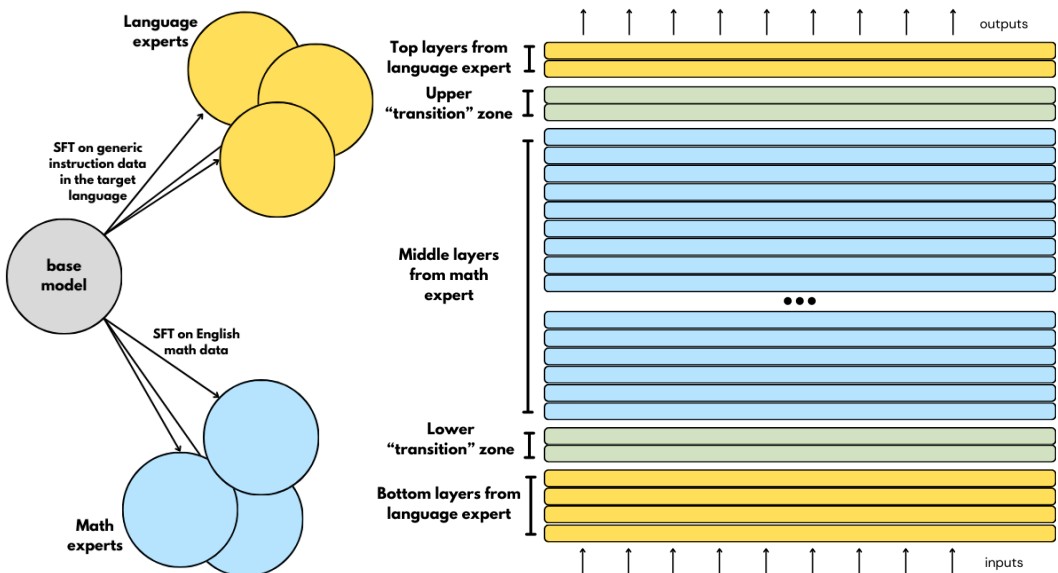

Figure 1: Our merging method which swaps in top and bottom transformer layers from a language expert into a math expert, buffered by a transition zone.

these variants, which we refer to as "experts", we combine their learned language and task capabilities by re-composing an LLM with a mix of parameters from each while avoiding negative interference. Notably, the top and bottom few transformer layers are selected from the language expert, and the middle transformer layers are selected from the math expert. The intuition behind this simple, yet strategic, method is informed by our analysis of the SFT updates that led to the experts. We find that the learned math capabilities are concentrated in the middle transformer layers, especially in the second half of the model. Meanwhile, we find the enhanced language skills to have come from parameters closest to the input and output, in line with previous literature stating that this is where the most language-specific representations are concentrated (Wendler et al., 2024; Alabi et al., 2024; Tang et al., 2024). Therefore, our methodology takes the most important layers from each expert and transfers math capabilities to the target language.

The resulting *layer-swapped* models deliver strong performance across numerous target languages (Swahili, Telugu, Bengali, Japanese) on MGSM (Shi et al., 2023), the manually-translated version of the Grade School Math benchmark (Cobbe et al., 2021). In all these major languages, where math SFT data is not readily available, *layer swapping* outperforms baselines, the individual experts, and model souping (Wortsman et al., 2022)) by 10% on average. For Swahili, *layer swapping* exceeds MGSM performance of models fine-tuned on a mixed Swahili and math SFT dataset when evaluated. Therefore, this methodology provides a simple and effective way to improve the capabilities of LLMs without the need for any task-specific data in the target language. In addition, it is inexpensive and fully post hoc, meaning that it has the potential to be practical in many settings. Fundamentally, the success of this method also provides empirical evidence for cross-lingual patterns in the latent structures of LLMs that can be further interpreted and exploited.

We discuss relevant literature in the following section. We then explain the analysis that led to our methodology and present *layer swapping* in detail in Sections 3 and 4. Next, we show empirical results and discuss findings in Sections 5 and 6. Finally, we propose future work prompted by the success of this methodology in Section 7.

## 2  RELATED WORK

### 2.1  MODEL MERGING

While the use of weight averaging to reduce noise in training well predates instability challenges in deep neural networks (Breiman, 1996), the use of model merging to combine trained model

checkpoints is an emerging research space in deep learning. Similar to ensembling model outputs (Dietterich, 2000), aggregating model weights also improves model robustness and generalization (Izmailov et al., 2018), even if trained on the same data (Ramé et al., 2022). Numerous studies seek to improve the pre-merging conditions, either via linearizing fine-tuning (Ortiz-Jimenez et al., 2023) or aligning weights (Ainsworth et al., 2023). Wortsman et al. (2022) develops an empirical method to average, or *soup*, model variants together to increase the search space during hyperparameter tuning. However, simple weight averaging, is vulnerable to negative transfer, or interference, between variants. To address this, methods have been proposed to selectively combine models at the individual weight level, using either the magnitude and direction of the fine-tuning deltas (Ilharco et al., 2023; Yadav et al., 2023; Davari & Belilovsky, 2024; Yu et al., 2024) or leveraging information theory (Matena & Raffel, 2022). In parallel, sparse fine-tuning methods have been developed to create fine-tuned models with a small proportion of weights changed (Guo et al., 2021; Sung et al., 2021; Xu et al., 2021), which then allows adding together fine-tuning updates with less overlap. Overall, model merging, especially inexpensive model souping, is very common in practice because it improves training stability, model robustness and generalization, and performance by increasing the search space or combining expertise in multi-task settings (Yang et al., 2024b).

## 2.2 LLM Multilinguality

The inability for a language model to learn more languages without undermining other capabilities—the *curse of multilinguality* (Conneau et al., 2020; Pfeiffer et al., 2022)—was a heavily studied problem in encoder models. Beyond increasing vocabulary capacity (Zheng et al., 2021; Liang et al., 2023), numerous works attempt to understand the quantity and location of language-specific parameters (Wang et al., 2020; Muller et al., 2021; Choenni et al., 2023) in multilingual encoders. While higher rates of language-specific parameters were found in the top and bottom layers (Chang et al., 2022; Choenni et al., 2024), language specialization occurs throughout the model. And while recently model scaling has mitigated the limitation from parameter quantity, the massive amount of labeled data needed to train LLMs presents a new challenge. In encoder models, cross-lingual transfer was enhanced by aligning cross-lingual representations (Ouyang et al., 2021; Patra et al., 2023; Gaschi et al., 2023), but this is difficult in decoder-only models (Jain et al., 2023). To boost transfer in post-training, several data augmentation solutions have been proposed, both for SFT (Qin et al., 2023; Chai et al., 2024) or reinforcement learning from human feedback (RLHF) (Dang et al., 2024; She et al., 2024; Lai et al., 2024). Recent work finds that English-centric LLMs process multilingual text by mapping them to and from English representations in the first and last transformer layers to take advantage of English-based capabilities (Kojima et al., 2024; Wendler et al., 2024; Tang et al., 2024; Alabi et al., 2024). This leads to benefits from prompting such LLMs to "think in English", thereby promoting this internal mapping (Shi et al., 2023; Zhang et al., 2024b).

## 2.3 Model Merging for Cross-Lingual Transfer

To address the limited representational capacity of multilingual models, many solutions have been proposed to strategically share or split parts of the model. This could be major blocks, as in mixture-of-experts (Fedus et al., 2022; NLLB et al., 2022), or a few parameters in each layer, as in cross-lingual adapters (Pfeiffer et al., 2020; 2022). Ansell et al. (2022) combines modular and sparse fine-tuning approaches in a two-stage SFT, where first task and language experts are fine-tuned and all weights that did not change more than a threshold are masked. Next, the experts are fine-tuned again from the pretrained model with this mask, creating sparse task vectors that can be composed together with lower rates of parameter overlap. This composable sparse fine-tuning was also adapted for larger decoder-only LLMs (Ansell et al., 2024). In this work, we develop a simpler and more flexible merging method for cross-lingual transfer that does not require fine-tuning more than once.

## 3 Preliminary Analysis

### 3.1 Setup

We start by training numerous math and language "experts" by fine-tuning LLAMA 3.1 8B (Llama et al., 2024). We perform SFT runs using next token prediction with 30-40k labeled samples with

varying hyperparameters[1] and for each type of expert, select the three best checkpoints. The math experts were fine-tuned on English math word problems from the Orca-Math synthetic dataset (Mitra et al., 2024). In order to select the three experts, we use results on the English splits of MGSM, as well as the average across languages. For the language experts, we select Swahili, Bengali, Telugu, and Japanese as the target languages. These are languages present in MGSM and other benchmarks discussed below, but where LLAMA 3.1's performance lags behind the top languages like Spanish. In addition, the lack of in-language math instruction data in these languages motivates the need for effective methods for *zero-shot* cross-lingual transfer. For each, we mix together samples from available instruction datasets in that language to create experts with enhanced general language and instruction-following capabilities. The resulting datasets contain many different types of tasks—such as translation, NER, and question-answering—but no math[2]. After numerous SFT runs on these "generic" multi-task datasets, the three checkpoints for each language are primarily selected based off their performance on the target language splits on BELEBELE (Bandarkar et al., 2024) and FLORES (NLLB et al., 2022). These simple tasks, reading comprehension and translation, are strong indicators of basic language understanding and generation capabilities. We also ensure slight improvement on MBPP (Austin et al., 2021), MMLU (Hendrycks et al., 2021), and MGSM as secondary measures of language improvement[3].

## 3.2 PARAMETER ANALYSIS OF SFT

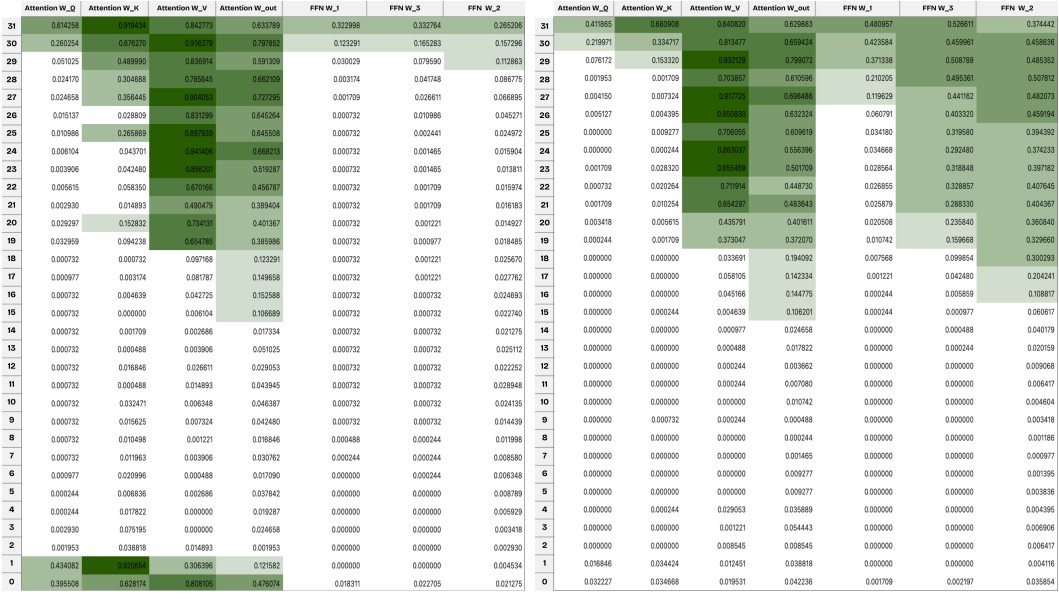

(2A) Japanese expert #1  (2B) Math expert #1

Figure 2: This visualization displays the location with more significant magnitude of change during SFT for two representative experts across transformer layers and parameter types. In detail, this shows the percentage of rows for each 2-dimensional parameter in the 32 transformer layers of LLAMA 3.1 8B where the mean absolute value is above a threshold ($1.9 \times 10^{-5}$ and $1.0 \times 10^{-5}$, respectively). Darker green shading represents parameters changing more significantly, relative to the others. Larger versions of these images, as well as for more experts, can be found in Appendix A.5

For these experts, we first investigate where the parameters are being updated the most and where they remain unchanged during fine-tuning. Using similar notation to Ilharco et al. (2023), let $\theta_{pre}, \theta_{ft}$ be the set of weights for the pretrained model and fine-tuned expert, respectively. We generalize the delta of fine-tuning as $\mathbf{W}_\Delta = \mathbf{W}_{ft} - \mathbf{W}_{pre}$ for all one- or two-dimensional parameters $\mathbf{W} \in \theta$. To compare deltas across parameter tensors with different shapes, we represent the

---

[1]Training specifics such as hyperparameters are provided in Appendix A.2
[2]Dataset details are provided in Appendix A.1
[3]See results of chosen experts in Appendix A.3

magnitude of change at the row level using the mean absolute value (MAV) of the difference $\mathbf{W}_\Delta$ (for one-dimensional parameters, this is simply the MAV across the vector).

We observe highly consistent patterns in the $\mathbf{W}_\Delta$ magnitudes among math experts and, separately, among language experts. The latter is particularly notable given the varying mixtures of tasks and data samples for each language. In Figure 2, we show a visualization for a representative language expert and a representative math expert. For the language expert, we find that the attention parameters (left four columns in the visualization, referring to $W_Q, W_K, W_V$, and $W_O$, respectively) are getting updated most significantly, notably in the first couple layers (bottom) and the last couple layers (top). The feed-forward layers (right three columns, referring to $W_1, W_3$, and $W_2$, respectively) do not change at all until the last few layers. In comparison, the math experts follow a different pattern. In these runs, the first half (bottom 16 layers) remain largely unchanged. In this second half, the attention parameters are being updated significantly, similar to language experts, yet the feed-forward layers are also getting changed quite a bit. To demonstrate the consistency across training runs and languages, we show the same visualization for more experts in Appendix A.5.

### 3.3 SPARSIFYING UPDATES

We then attempt to create *sparse* model updates for increased composability similar to Ansell et al. (2022), but without retraining. We attempt to utilize the magnitude of the deltas to selectively update the model without undermining performance gains. Concretely, we use thresholds to determine whether to apply the update or revert the value to the original value in the pretrained LLAMA 3.1. This is done at row-level granularity in order to not partially modify linear transformations. However, in our analysis across the math experts, we find that more than 70-80% of model parameters are required to be updated for the increase in performance to remain equivalent. Such a small rate of sparsification would not significantly reduce interference between the math and language experts.

We next attempt location-based sparsification. This means leveraging our intuition of what patterns occur during fine-tuning to select specific parameter tensors to merge or not. A breakthrough was made when we revert the first five and last two transformer layers of our math experts to their original values. Despite undoing the SFT updates, the math experts maintain their increased performance. This is significant when considered alongside our intuition that the first few and last few transformer layers contain the most important language-specific parameters. This is based on our above analysis and is supported by previous multilingual interpretability research on both encoder-decoder models (Chang et al., 2022; Choenni et al., 2024) and decoder-only models (Tang et al., 2024; Zhang et al., 2024c). Our discovery that SFT updates to these same layers are not critical for mathematical reasoning led us to experiment with swapping them in from the language expert.

## 4 METHODOLOGY

### 4.1 LAYER SWAPPING

The *layer swapping* methodology takes two experts, one fine-tuned on the target language and the other on the target task in English, and re-composes a single model with the top and bottom transformer layers from the language expert and the middle from the math expert.

As displayed in Figure 1, we additionally design a *transition zone* in between the off-the-shelf layers from each expert. These buffer layers are weighted averages of the respective layers from each expert that ensure that the output of one layer does not directly input into a layer fine-tuned separately. While intuitively these transition zones seem necessary, we do not find empirical evidence that they provide statistically significant benefit over replacing them with the math expert's layers. However, as discussed in Section 3.1, our experts were not fine-tuned for very long and therefore the latent representation spaces after each layer would not have diverged very much from each other. This explains, in theory, why transition zones were not helpful in our setting. We conjecture that if the experts had been trained further (or simply with higher learning rates), such a buffer zone would be necessary. We therefore still present it as a central component of our *layer swapping* methodology.

The implementation of *layer swapping* is as simple as iterating through the state dictionary of the math expert and for each parameter, either: (1) keeping it as is, (2) replacing its value with that of the language expert, or (3) averaging that of the math and language expert (see Algorithm 1).

---

**Algorithm 1** *Layer Swapping*

---

**Input:**    task expert $\theta_{task}$, language expert $\theta_{lang}$, lower layers to swap $b$, upper layers to swap $u$, lower transition layers $t_b$, upper transition layers $t_u$, weight of each expert $\alpha_{task}, \alpha_{lang}$, number of model layers $L$

**Output:**    Merged model $\theta_{merged}$

1: **for** parameter name $n$ in models parameters **do**
2:    $l \leftarrow$ layer number of $n$, N/A if $n$ not attention or feedforward parameters
3:    **if** $l < b$ or $l > L - 1 - u$ **then**
4:        $\theta_{merged}\{n\} \leftarrow \theta_{lang}\{n\}$             # top & bottom layers from language expert
5:    **else if** $l > b - 1 + t_b$ or $l < L - u - t_u$ **then**
6:        $\theta_{merged}\{n\} \leftarrow \theta_{task}\{n\}$             # middle layers from task expert
7:    **else**
8:        $\theta_{merged}\{n\} \leftarrow (\alpha_{task} * \theta_{task}\{n\} + \alpha_{lang} * \theta_{lang}\{n\})/(\alpha_{task} + \alpha_{lang})$  # transition zone
9:    **end if**
10: **end for**
11: Return $\theta_{merged}$

---

## 4.2 CONFIGURATION

*Layer swapping* has several components that can be configured in various ways. Most notably, the number of layers to swap at the top and bottom and the number of layers to include in the respective transition zones. We tested how to configure these swapped layers and transition zones to most effectively merge these models, empirically. We find, however, that there is a wide range of possible configurations in which this methodology is still very effective. Note that all these experiments were conducted on a 32-layer LLAMA 3.1 transformer model. For a model of this size, we find that the desired configuration of each component is in the ranges listed below, with our default values underlined:

1. The number of bottom layers to swap $b \in \{3, 4, \underline{5}\}$

2. The number of top layers to swap $u \in \{0, 1, \underline{2}\}$

3. The number of layers in the lower transition zone $t_b \in \{\underline{0}, 1, 2\}$

4. The number of layers in the upper transition zone $t_u \in \{\underline{0}, 1, 2\}$

5. The transition zones are averages of the layers from both (i.e. soups) that can be unweighted or magnitude-adjusted weighted averages $(\alpha_{task}, \alpha_{lang})$.

6. The non-transformer parameters (input token embeddings, output layers, etc.) work best if they are also averages of the two experts, as opposed to simply from the language expert.

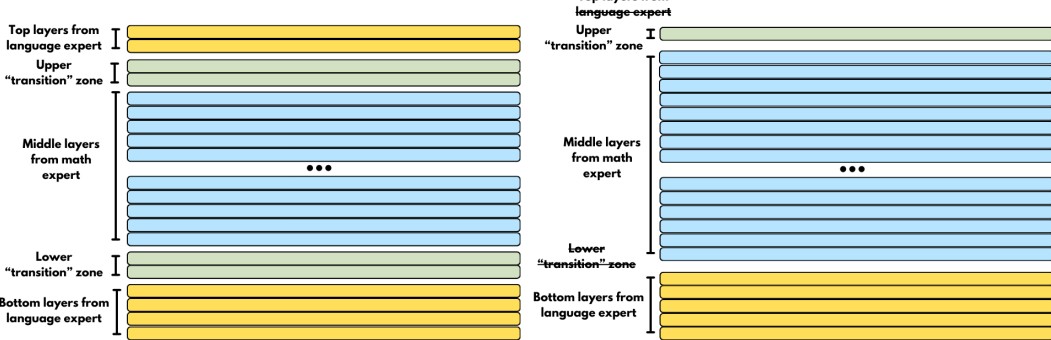

(3A) Possible config. with more layers swapped in.      (3B) Possible config. with less layers swapped in.

Figure 3: The comparison of the maximum (left) and minimum (right) swapping setups that we find effective empirically. Note on the right, there are no upper layers directly from the language expert.

Although the results were largely equivalent within this range, some patterns did exist. We find that for languages where performance was lower (e.g. Telugu), the higher performing configurations typ-

ically had more layers from the language expert. This is perhaps because the boost from improving the language capabilities was more important relative to languages LLAMA 3.1 was already better at. For example for languages such as Japanese, the best configuration has close to the minimum layers from language expert, similar to the minimum swap illustrated in Figure 3. The math expert where the most cross-lingual transfer occurred naturally during SFT (that is, with the highest scores on all four target languages), named "math expert #2", performed understandably better with fewer layers from the language expert swapped in. Generally, we conjecture that lower-resource languages tend to require more layers from the language expert.

## 5 EMPIRICAL RESULTS

### 5.1 EXPERIMENTS ON SWAHILI, TELUGU, JAPANESE, AND BENGALI

As discussed in Section 3.1, we launched SFT runs with different hyperparameters and on five different datasets: math data in English and generic instruction data in Bengali, Swahili, Telugu, and Japanese. For each dataset, we selected three checkpoints, striking a balance between top performance and sufficient diversity to ensure robust experimental results. For each language, the three language experts could pair with any of the three math experts, which gives nine merged models to evaluate. To understand the effectiveness of *layer swapping*, we present the MGSM scores of the individual experts and the base model, LLAMA 3.1, to determine whether *layer swapping* constructively combines the learned task and capabilities. In addition, we present results from the classic model soup (Wortsman et al., 2022). Model souping is the most common model merging method in practice, used in pre- and post-training for purposes such as boosting model robustness and generalization, but also to combine capabilities of experts (Yu et al., 2024). In vanilla model souping, the merged parameters are simple averages of the parameters of the input experts. For each language, we additionally evaluate adjusting for the magnitude of change from the base model to the expert using a weighted average. Using notation from Section 3.2, the resulting weight of each expert is the inverse of the average MAV for all rows in all $\mathbf{W}_\Delta$ for that expert. For all model soup results presented, we select the highest MGSM score amongst these possible configurations. Given the number of experts and merged pairs in our evaluations, we present two aggregate metrics: the mean and maximum of evaluations. While the mean demonstrates the consistency of each method, the maximum demonstrates the ceiling of the method, which is often more desirable given the ability to iterate through numerous settings during model training.

Table 1: **MGSM 8-shot results of *layer swapping* across four languages** compared to the individual experts and model souping. Note that for aggregate statistics of the individual SFT runs, we select the 3 best checkpoints from numerous training runs with periodic checkpointing. The merging methods are aggregated over the 9 pairs (3 language experts $\times$ 3 math experts), which means the min, avg, and max measures are not perfectly comparable.

| Setting | LLAMA 3.1 8B | language expert | math expert | model soup | *layer swap* | *layer swap* |
|---|---|---|---|---|---|---|
| **Details** | | top 3 training runs | top 3 training runs | best config, 9 pairs | default config, 9 pairs | best config, 9 pairs |
| **Swahili avg** | *24.8* | 24.7 | 29.5 | 29.3 | 32.4 | **32.8** |
| **Swahili max** | *24.8* | 25.6 | 32.8 | 32.0 | 36.4 | **37.2** |
| **Telugu avg** | *12.0* | 20.0 | 20.1 | 20.9 | 22.7 | **23.0** |
| **Telugu max** | *12.0* | 22.4 | 24.0 | 26.4 | 27.6 | **27.6** |
| **Bengali avg** | *29.2* | 33.5 | 38.3 | 36.8 | 37.1 | **38.7** |
| **Bengali max** | *29.2* | 35.2 | 44.4 | 38.4 | 40.4 | **45.2** |
| **Japanese avg** | *33.6* | 35.9 | **42.7** | 38.7 | 38.5 | 40.1 |
| **Japanese max** | *33.6* | 36.8 | **44.8** | 40.0 | 40.8 | 43.2 |

As displayed in Table 1, we find that *layer swapping* consistently outperforms these baselines on MGSM. For Swahili and Telugu, the default *layer swapping* configuration scores higher than both the individual experts across all 9 pairs. This provides tremendous evidence that this merging method

prevents negative interference. For Bengali, where the math expert already performs significantly better than for Swahili and Telugu, numerous layer-swapping configurations consistently surpass the math expert's average score. In addition, the best *layer swapped* model performs higher than the best math expert on its own. For Japanese, here we find a lower average performance compared to the individual math experts. However, we note that our Japanese experts were perhaps the weakest; the increases in performance across BELEBELE, FLORES, MBPP, and MMLU after SFT were minimal and with the data and hyperparameters tested, we were unable to do better. Furthermore, MGSM performance of both the base LLAMA 3.1 and the math experts was already decent in Japanese, prior to merging. Compared to model souping, *layer swapping* consistently outperforms it in terms of both maximum and average performance in all languages. The model soup results typically lie between the individual results of the experts.

We further evaluated these layer-swapped models on BELEBELE, FLORES, MBPP, and MMLU to ensure no over-optimization to the MGSM benchmark and find that the results are on par, usually even higher, than the base model and the individual experts. In addition, we manually checked a sample of text generations to further ensure the quality of the resulting model. Given that there are several language and math experts, we evaluate combining *layer swapping* with model souping. We do so by souping the three experts for each of the languages before *layer swapping* with a soup of the three math experts and present results in Table 6 in the Appendix. We find that these language and math soups perform on par with the individual pairs when *layer swapped* together. The feasibility in these settings shows that *layer swapping* can further extend the search space for fine-tuned models.

Given these results, we conjecture that perhaps *layer swapping* provides the biggest benefit for lower-resource languages (e.g. Swahili, Telugu), although such a conclusion would require evaluating more than four languages. This conjecture would be explained by the difficulty of improving language capabilities with low-data SFT when LLAMA 3.1 was already pretrained on more tokens in that language. Additionally, for higher-resource languages, cross-lingual transfer occurs more naturally when fine-tuning on English math data.

## 5.2 FURTHER EVALUATIONS ON SWAHILI

Given that here, model souping appears to be susceptible to negative interference when merging, we additionally evaluate TIES-merging (Yadav et al., 2023) for Swahili. This merging method resolves interference between experts by sequentially trimming conflicting values, determining the sign of each weight, and then merging the resulting weights. In addition, we present an alternative for Swahili where, instead of post hoc model merging, we mix the language and math datasets and do a single fine-tuning run with samples from both. Instead of training two experts, each on 30-40k samples, the joint SFT is over the union (80k samples). Identical to the training of experts, we launch many SFT runs with different hyperparameters and select the best three checkpoints.

Table 2: **MGSM 8-shot results of *layer swapping* for Swahili** in more detail and with two additional comparisons, TIES-merging and dataset merging. We display the minimum performance in Swahili, as well as the average across all 9 languages in MGSM and in English.

| Setting | LLAMA 3.1 8B | Swahili expert | math expert | swh&math joint SFT | model soup | TIES-merging | *layer swap* | *layer swap* |
|---|---|---|---|---|---|---|---|---|
| **Details** | | top 3 training runs | top 3 training runs | top 3 training runs | best config, 9 pairs | best config, 9 pairs | default config, 9 pairs | best config, 9 pairs |
| **Swahili min** | *24.8* | 23.6 | 27.2 | 31.6 | 25.6 | 25.2 | 29.6 | 29.2 |
| **Swahili avg** | *24.8* | 24.7 | 29.5 | 32.1 | 29.3 | 29.5 | 32.4 | **32.8** |
| **Swahili max** | *24.8* | 25.6 | 32.8 | 32.8 | 32.0 | 32.4 | 36.4 | **37.2** |
| **English avg** | *56.0* | 55.7 | 66.2 | 64.3 | 62.0 | 60.1 | 64.7 | 64.4 |
| **All langs avg** | *37.7* | 37.5 | 45.4 | 46.0 | 43.0 | 41.8 | 44.1 | 44.4 |

Table 2 displays these more detailed results for Swahili. We find that TIES-merging, similar to model souping, consistently underperforms compared to *layer swapping*. For mixing the Swahili and math data prior to fine-tuning, we are able to achieve results comparable to *layer swapping* on average, with the average MGSM score for this joint SFT being less than one point lower. How-

ever, the maximum performance of these checkpoints lagged the best *layer-swapped* pairs by 4.4 percentage points. This means that the ceiling for cross-lingual transfer is significantly higher with this methodology than simply mixing datasets together. This is significant because our method of merging two variants fine-tuned on separate datasets proves to be more effective than an extended fine-tuning on the combined datasets, consistent with Aakanksha et al. (2024).

We note that by swapping in layers from the Swahili expert, MGSM performance in English and other non-Swahili languages decreases from the math expert. This is expected given that we optimize for a different language, yet the decrease is relatively small nevertheless. For comparison, we see that performance drops much more for the other merging methods on these non-target languages.

## 6 DISCUSSION

*Layer swapping* is a simple, yet effective method for merging models for cross-lingual transfer. The success of this post hoc method prompts numerous key insights.

In settings where in-language task data is unavailable or rare, such as labeled math samples in Telugu, *layer swapping* provides a very effective way to create a capable model with simply English task data and general target language data. For lower-resource languages, such a constrained scenario is extremely common in practice, especially when it comes to instruction fine-tuning data. All other baselines that we evaluate in such a scenario are not able to combine language and math capabilities as consistently and effectively, as discussed in Section 5.

The surprising effectiveness of this method can be explained by the observation that English-centric LLMs do most of their internal thinking in English, which aligns with recent literature (Wendler et al., 2024; Zhang et al., 2024b; Tang et al., 2024; Kojima et al., 2024). Therefore, our method exploits the finding that the most important language-specific parameters are those in the top and bottom transformer layers that map multilingual input to and from English representations. Potentially, we can reinterpret the functionality of these transformer layers as, to a certain degree, natural language *interfaces* to broader model intelligence. Our method preserves the updates to the middle layers, as we hypothesize that this is where most of the improvements to the LLM's mathematical reasoning are concentrated. Consequently, this separability of multilingual parameters from other model capabilities could motivate a new generation of modular solutions for multilingualism in decoder-only LLMs—akin to MoE or adapter-based approaches designed for encoder models (Pfeiffer et al., 2020; 2022; Liu et al., 2023). Regardless, this work suggests that further interpretability research into LLM multilinguality could lead to more cost-effective solutions to boost non-English capabilities.

Model souping is popular because of its flexibility, convenience, and ability to expand the model search space. In this regard, *layer swapping* provides a more effective alternative for the multilingual setting with the same advantages. This method can be implemented fully post hoc between any number of checkpoints using simple parameter arithmetic, with no further fine-tuning. Because of how inexpensive it is to implement, it enables the ability to quickly iterate through and test many configurations (e.g. the number of swapped transformer layers). Similarly to souping, the simplicity of this method means that it has the potential to work at any stage of training (pretraining, fine-tuning, preference tuning, etc.). In multi-task fine-tuning scenarios, this could allow for non-English capabilities being boosted on the side, separately from other *reasoning* capabilities—such as math, multi-hop reasoning, coding, safety, etc.—and then models are merged back together post hoc to combine the learned skills.

In terms of model merging, our analysis indicates that weight-level techniques are perhaps not as effective in reducing interference when merging parameters together, as compared to parameter-level or even layer-level merges. A possible explanation is that modifying individual weight values may disrupt the linear dependence within the transformations defined by the weight tensors. Conversely, the recent success of self-speculative decoding in LLMs (Zhang et al., 2024a; Elhoushi et al., 2024) suggests that the token representation spaces of most layers across the model are very similar. If so, keeping entire layers in tact may alleviate the risk of undermining newfound learning. Another reason could be that using very granular magnitude measures may be too noisy to properly indicate which weight updates are important and which are not.

## 7 LIMITATIONS AND FUTURE WORK

The positive results of this *layer swapping* methodology largely raise many questions about the extent to which it is effective and in what settings. We propose here many further investigations that will build a broader understanding of the method's practical scope.

*Freezing parameters before training experts*: Instead of re-composing a model with layers from separate models, a logical next evaluation would be to freeze model layers from the beginning. Intuitively, this would only be more effective from a performance standpoint, as there is no ad hoc merging of experts from disparate training jobs. However, a significant benefit of our solution is that it is all post hoc and flexible. Parameter freezing would require knowing which configuration would work prior to training or require numerous training runs to find the best configuration.

*LLMs with different pretraining*: An explanation for why this method works so well is that English-centric LLMs "think in English", which is shown in more than Llama 3.1. It must be studied whether LLMs with balanced multilingual pretraining have similarly consolidated representations, and if not, whether *layer swapping* breaks.

*Different model sizes*: LLMs with more parameters, either from more transformer layers or from larger hidden dimensions, would clearly require new merging configurations. In addition, it remains to be seen how related attributes such as sparsity, redundancy, and the quantity of cross-lingual shared parameters, would influence the performance of *layer swapping*.

*Lower-resource langs*: Among the limited languages evaluated, the lower-resource ones benefited most from *layer swapping*. However, the model needs minimum capabilities in the target language for zero-shot cross-lingual transfer. It is unclear how well the LLM needs to understand the target language for this to be effective.

*Other reasoning tasks*: This work only tackles mathematical reasoning, but it is conceivable that other complex reasoning capabilities are concentrated in parameters separate from language capabilities in a similar manner. That being said, the configuration required for *layer swapping* may be completely different and may require a preliminary analysis as in Section 3.

*Parameter-efficient fine-tuning*: Since *layer swapping* treats transformer layers as unified blocks, it would be equivalent if the model were fine-tuned using parameter-efficient SFT methods such as LoRA, which consists of inserting adapter weights into the transformer layer (Hu et al., 2022). Modifying other model merging methods for such adapters is simple (Zhang et al., 2023), and therefore *layer swapping* has the potential to be effective in parameter-efficient fine-tuning settings.

*Different training stage*: We limit the focus of this work on low-data SFT. Model souping, itself, is implemented in practice at all stages (e.g. pretraining, CPT), and *layer swapping* has the potential to be effective there as well. Similarly, model souping is effective even when the checkpoints have been fine-tuned significantly. It is unclear what would be the point for our methodology where the experts have diverged too significantly that they would no longer recombine well. Either way, it could still enable a multi-step training where experts are iteratively fine-tuned and merged successively, analogous to gradient aggregation in data-distributed training.

## 8 CONCLUSION

In this paper, we address the difficulty of training LLMs to perform tasks in languages where labeled task-specific data does not exist. We first trained separate experts on English math data and generic instruction data in numerous target languages. An analysis of the importance of different parameter updates during fine-tuning led to the development of the *layer swapping* method which swaps in the top and bottom layers from language experts into the math expert. Surprisingly, this simple model merging provides one of the highest performing methods to fine-tune an LLM for math in a target language without the presence of target language data. The strong intuition behind this solution and its flexibility, simplicity, and effectiveness provide vast potential to be practical in many other settings. This method can be potentially adapted for new pretrained models, target tasks, target languages, training stages, training setups, and more. In addition, this method indicates better model interpretability of multilinguality can lead to more efficient methods for transferring English capabilities to lower resource languages.

REPRODUCIBILITY

To reproduce the fine-tuning of our "expert" models, we describe the process in Section 3.1 and provide further details on the data and fine-tuning hyperparameters in Appendix A.1 and A.2. Further details for our analysis described in Section 3.2 can be found in Appendix A.5. We provide pseudocode of the *layer swapping* algorithm defined in Section 4 in Algorithm 1. All of our experiments for which results are presented are thoroughly explained in Section 5.

ACKNOWLEDGMENTS

The authors acknowledge the crucial support provided by Mostafa Elhoushi, Tanmay Parekh, Jiabao Ji, and Chloe Bi that made this project possible.

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

# A APPENDIX

## A.1 FINE-TUNING DATASETS

Table 3: Datasets used for supervised-fine-tuning (SFT) in this project

| Category | Datasets | URL |
|---|---|---|
| Math | Orca Math word problems dataset from Microsoft (Mitra et al., 2024) | `https://huggingface.co/datasets/microsoft/orca-math-word-problems-200k` |
| Telugu | Aya Dataset from Cohere for AI (Singh et al., 2024a) | `https://huggingface.co/datasets/CohereForAI/aya_dataset` |
| | NLLB English-Telugu translation data from FAIR (NLLB et al., 2022) | `https://huggingface.co/datasets/allenai/nllb` |
| | English instruction dataset, machine translated to Telugu | |
| Bengali | Aya Dataset by Cohere for AI (Singh et al., 2024a) | `https://huggingface.co/datasets/CohereForAI/aya_dataset` |
| | English-Bengali translation data from NLLB (NLLB et al., 2022) | `https://huggingface.co/datasets/allenai/nllb` |
| | IndicShareLlama dataset from AI4Bharat (Khan et al., 2024) | `https://huggingface.co/datasets/ai4bharat/indic-align` |
| | BongChat dataset from Lumatic AI | `https://huggingface.co/datasets/lumatic-ai/BongChat-v1-253k` |
| Swahili | Aya Dataset by Cohere for AI (Singh et al., 2024a) | `https://huggingface.co/datasets/CohereForAI/aya_dataset` |
| | English-Swahili translation data from NLLB (NLLB et al., 2022) | `https://huggingface.co/datasets/allenai/nllb` |
| | Inkuba dataset from Lelapa (Tonja et al., 2024) | `https://huggingface.co/datasets/lelapa/Inkuba-instruct` |
| | xP3 MT dataset from BigScience, with FLoRES samples removed (Muennighoff et al., 2022) | `https://huggingface.co/datasets/bigscience/xP3mt` |
| Japanese | Aya Dataset by Cohere for AI (Singh et al., 2024a) | `https://huggingface.co/datasets/CohereForAI/aya_dataset` |
| | English-Japanese translation data from NLLB (NLLB et al., 2022) | `https://huggingface.co/datasets/allenai/nllb` |
| | LLM-Japanese dataset from Izumi Lab (Hirano et al., 2023) | `https://huggingface.co/datasets/izumi-lab/llm-japanese-dataset` |
| | Ichikara dataset from RIKEN AIP | `https://huggingface.co/datasets/p1atdev/ichikara-instruction` |
| | Dolly dataset from Databricks, machine translated to Japanese | `https://huggingface.co/datasets/kunishou/databricks-dolly-15k-ja` |

## A.2 SUPERVISED FINE-TUNING DETAILS

While many hyperparameters were tried, below is listed the hyperparameter configurations that led to the three best checkpoints (the "experts") for each category. Note that in all runs, we do checkpointing every 5000 samples and use a different random seed for data sampling.

Table 4: Hyperparameters for the training runs that led to each of our "experts"

| Expert | Learn Rate | Batch Size | Seq. Length | weight decay | clip, max norm | sched. | warmup steps | $\beta2$ |
|---|---|---|---|---|---|---|---|---|
| math #1 | $2.0 \times 10^{-8}$ | 4 | 2048 | 0.1 | 1.0 | Linear | 1000 | 0.99 |
| math #2 | $1.0 \times 10^{-7}$ | 4 | 2048 | 0.01 | 0.5 | Linear | 1000 | 0.99 |
| math #3 | $4.0 \times 10^{-8}$ | 4 | 2048 | 0.1 | 1.0 | Linear | 500 | 0.999 |
| jpn #1 | $1.0 \times 10^{-7}$ | 8 | 1024 | 0.1 | 0.5 | WSD | 1000 | 0.995 |
| jpn #2 | $1.0 \times 10^{-7}$ | 8 | 1024 | 0.1 | 0.5 | WSD | 1000 | 0.99 |
| jpn #3 | $2.0 \times 10^{-7}$ | 8 | 1024 | 0.01 | 0.5 | WSD | 1000 | 0.99 |
| swh #1 | $7.0 \times 10^{-8}$ | 8 | 1024 | 0.1 | 0.5 | WSD | 1000 | 0.99 |
| swh #2 | $2.0 \times 10^{-7}$ | 8 | 1024 | 0.05 | 0.5 | WSD | 1000 | 0.99 |
| swh #3 | $1.0 \times 10^{-7}$ | 8 | 1024 | 0.1 | 1.0 | WSD | 1000 | 0.999 |
| tel #1 | $7.0 \times 10^{-8}$ | 8 | 1024 | 0.1 | 0.5 | WSD | 1000 | 0.99 |
| tel #2 | $5.0 \times 10^{-8}$ | 8 | 1024 | 0.05 | 0.5 | WSD | 300 | 0.99 |
| tel #3 | $1.0 \times 10^{-7}$ | 4 | 2048 | 0.1 | 0.5 | WSD | 1000 | 0.99 |
| ben #1 | $7.0 \times 10^{-8}$ | 8 | 1024 | 0.1 | 0.5 | WSD | 1000 | 0.99 |
| ben #2 | $5.0 \times 10^{-8}$ | 8 | 1024 | 0.05 | 0.5 | WSD | 300 | 0.99 |
| ben #3 | $1.0 \times 10^{-7}$ | 4 | 2048 | 0.1 | 0.5 | WSD | 1000 | 0.99 |

## A.3 PERFORMANCE OF INDIVIDUAL EXPERTS

Table 5: Comprehensive results for each of the set of experts on tasks in the target language. Each number represents an average over the 3 selected experts. The "10-lang avg" column presents the results across 10 languages (English, German, French, simplified Chinese, Russian, Spanish, Japanese, Bengali, Swahili, Telugu).

| Task | Language | LLAMA 3.1 8b | Average across 3-experts | | | | |
|---|---|---|---|---|---|---|---|
| | | | Math | Japanese | Bengali | Swahili | Telugu |
| **MGSM** | English | 0.560 | 0.663 | 0.561 | 0.563 | 0.557 | 0.561 |
| em@maj1, 8-shot | Japanese | 0.336 | 0.427 | 0.359 | | | |
| | Bengali | 0.292 | 0.383 | | 0.335 | | |
| | Swahili | 0.248 | 0.295 | | | 0.247 | |
| | Telugu | 0.120 | 0.201 | | | | 0.200 |
| | 10-lang avg | 0.311 | 0.454 | | | | |
| **BELEBELE results** | English | 0.871 | 0.877 | | | | |
| accuracy, 5-shot | Japanese | 0.760 | 0.754 | 0.766 | | | |
| | Bengali | 0.649 | 0.654 | | 0.683 | | |
| | Swahili | 0.553 | 0.557 | | | 0.584 | |
| | Telugu | 0.546 | 0.557 | | | | 0.602 |
| **FLORES results** | Eng-Jpn | 20.6 | 20.5 | 20.5 | | | |
| BLEU, 0-shot | Jpn-Eng | 28.5 | 28.5 | 28.9 | | | |
| | Eng-Ben | 18.8 | 18.8 | | 17.9 | | |
| | Ben-Eng | 30.5 | 29.7 | | 29.0 | | |
| | Eng-Swh | 12.5 | 12.9 | | | 13.0 | |
| | Swh-Eng | 31.5 | 31.5 | | | 30.2 | |
| | Eng-Tel | 15.7 | 15.2 | | | | 16.5 |
| | Tel-Eng | 29.2 | 29.0 | | | | 27.6 |
| **MBPP result** | Japanese | 0.464 | 0.470 | 0.462 | | | |
| pass @ 1, 0-shot | Bengali | 0.438 | 0.436 | | 0.462 | | |
| | Swahili | 0.402 | 0.407 | | | 0.407 | |
| | Telugu | 0.410 | 0.412 | | | | 0.437 |
| **MMLU result** | Japanese | 0.520 | 0.518 | 0.518 | | | |
| accuracy, 5-shot | Bengali | 0.422 | 0.420 | | 0.436 | | |
| | Swahili | 0.419 | 0.420 | | | 0.425 | |
| | Telugu | 0.415 | 0.412 | | | | 0.427 |

## A.4 RESULTS FROM COMBINING WITH SOUPING

Table 6: MGSM 8-shot results when combining *layer swapping* with model souping of the 3 experts for each category. "Multilingual soup" refers to the uniform soup of all 12 language experts.

| Setting | model soup, pairwise | *layer swap,* pairwise | *layer swap,* language soup w/ math soup | *layer swap,* multilingual soup w/ math soup |
|---|---|---|---|---|
| Details | default config, avg of 9 pairs | default config, avg of 9 pairs | default config, 1 version | default config, 1 version |
| Swahili | 29.3 | 32.4 | **32.8** | 29.6 |
| Telugu | 20.9 | **22.7** | 22.0 | 21.2 |
| Bengali | 36.8 | 37.1 | **40.4** | 36.0 |
| Japanese | 38.7 | 38.5 | 39.2 | **41.6** |

## A.5 PARAMETER-LEVEL VISUALIZATIONS OF THE EXPERT VECTORS

Larger parameter-level visualizations for a number of the experts, each visualization is the same as described in Figure 2

| | attention.wq.weight | attention.wk.weight | attention.wv.weight | attention.wo.weight | feed_forward.w1.weight | feed_forward.w3.weight | feed_forward.w2.weight |
|---|---|---|---|---|---|---|---|
| 31 | 0.411865 | 0.680908 | 0.840820 | 0.629883 | 0.480957 | 0.526611 | 0.374442 |
| 30 | 0.219971 | 0.334717 | 0.813477 | 0.659424 | 0.423584 | 0.459961 | 0.458636 |
| 29 | 0.076172 | 0.153320 | 0.932129 | 0.799072 | 0.371338 | 0.508789 | 0.485352 |
| 28 | 0.001953 | 0.001709 | 0.703857 | 0.610596 | 0.210205 | 0.495361 | 0.507812 |
| 27 | 0.004150 | 0.007324 | 0.917725 | 0.698486 | 0.119629 | 0.441162 | 0.482073 |
| 26 | 0.005127 | 0.004395 | 0.850830 | 0.632324 | 0.060791 | 0.403320 | 0.459194 |
| 25 | 0.000000 | 0.009277 | 0.706055 | 0.609619 | 0.034180 | 0.319580 | 0.394392 |
| 24 | 0.000000 | 0.000244 | 0.863037 | 0.556396 | 0.034668 | 0.292480 | 0.374233 |
| 23 | 0.001709 | 0.028320 | 0.855469 | 0.501709 | 0.028564 | 0.318848 | 0.397182 |
| 22 | 0.000732 | 0.020264 | 0.711914 | 0.448730 | 0.026855 | 0.328857 | 0.407645 |
| 21 | 0.001709 | 0.010254 | 0.654297 | 0.483643 | 0.025879 | 0.288330 | 0.404367 |
| 20 | 0.003418 | 0.005615 | 0.435791 | 0.401611 | 0.020508 | 0.235840 | 0.360840 |
| 19 | 0.000244 | 0.001709 | 0.373047 | 0.372070 | 0.010742 | 0.159668 | 0.329660 |
| 18 | 0.000000 | 0.000000 | 0.033691 | 0.194092 | 0.007568 | 0.099854 | 0.300293 |
| 17 | 0.000000 | 0.000000 | 0.058105 | 0.142334 | 0.001221 | 0.042480 | 0.204241 |
| 16 | 0.000000 | 0.000000 | 0.045166 | 0.144775 | 0.000244 | 0.005859 | 0.108817 |
| 15 | 0.000000 | 0.000244 | 0.004639 | 0.106201 | 0.000244 | 0.000977 | 0.060617 |
| 14 | 0.000000 | 0.000000 | 0.000977 | 0.024658 | 0.000000 | 0.000488 | 0.040179 |
| 13 | 0.000000 | 0.000000 | 0.000488 | 0.017822 | 0.000000 | 0.000244 | 0.020159 |
| 12 | 0.000000 | 0.000000 | 0.000244 | 0.003662 | 0.000000 | 0.000000 | 0.009068 |
| 11 | 0.000000 | 0.000000 | 0.000244 | 0.007080 | 0.000000 | 0.000000 | 0.006417 |
| 10 | 0.000000 | 0.000000 | 0.000000 | 0.010742 | 0.000000 | 0.000000 | 0.004604 |
| 9 | 0.000000 | 0.000732 | 0.000244 | 0.000488 | 0.000000 | 0.000000 | 0.003418 |
| 8 | 0.000000 | 0.000000 | 0.000000 | 0.000244 | 0.000000 | 0.000000 | 0.001186 |
| 7 | 0.000000 | 0.000000 | 0.000000 | 0.001465 | 0.000000 | 0.000000 | 0.000977 |
| 6 | 0.000000 | 0.000000 | 0.000000 | 0.009277 | 0.000000 | 0.000000 | 0.001395 |
| 5 | 0.000000 | 0.000000 | 0.000000 | 0.009277 | 0.000000 | 0.000000 | 0.003836 |
| 4 | 0.000000 | 0.000244 | 0.029053 | 0.035889 | 0.000000 | 0.000000 | 0.004395 |
| 3 | 0.000000 | 0.000000 | 0.001221 | 0.054443 | 0.000000 | 0.000000 | 0.006906 |
| 2 | 0.000000 | 0.000000 | 0.008545 | 0.008545 | 0.000000 | 0.000000 | 0.006417 |
| 1 | 0.016846 | 0.034424 | 0.012451 | 0.038818 | 0.000000 | 0.000000 | 0.004116 |
| 0 | 0.032227 | 0.034668 | 0.019531 | 0.042236 | 0.001709 | 0.002197 | 0.035854 |

Figure 4: Visualization of the magnitude of change during SFT for "math expert #1"

| | attention.wq.weight | attention.wk.weight | attention.wv.weight | attention.wo.weight | feed_forward.w1.weight | feed_forward.w3.weight | feed_forward.w2.weight |
|---|---|---|---|---|---|---|---|
| 31 | 0.539307 | 0.781738 | 0.635742 | 0.425293 | 0.637451 | 0.687256 | 0.351214 |
| 30 | 0.373047 | 0.511963 | 0.749512 | 0.559082 | 0.533203 | 0.479004 | 0.322126 |
| 29 | 0.207520 | 0.455322 | 0.818604 | 0.490234 | 0.436279 | 0.428711 | 0.346959 |
| 28 | 0.024170 | 0.059570 | 0.458740 | 0.375977 | 0.248047 | 0.346924 | 0.323730 |
| 27 | 0.025391 | 0.208252 | 0.785156 | 0.470215 | 0.121338 | 0.252686 | 0.276437 |
| 26 | 0.019043 | 0.074951 | 0.517578 | 0.384277 | 0.056641 | 0.182617 | 0.234375 |
| 25 | 0.006104 | 0.104736 | 0.458496 | 0.353271 | 0.021484 | 0.108887 | 0.179478 |
| 24 | 0.007080 | 0.016113 | 0.649414 | 0.354492 | 0.014893 | 0.083008 | 0.151228 |
| 23 | 0.008301 | 0.083008 | 0.650391 | 0.321777 | 0.010498 | 0.084961 | 0.152134 |
| 22 | 0.012207 | 0.114502 | 0.420410 | 0.280518 | 0.012695 | 0.100342 | 0.170898 |
| 21 | 0.053467 | 0.168457 | 0.276123 | 0.305664 | 0.012451 | 0.102295 | 0.172642 |
| 20 | 0.106445 | 0.091064 | 0.258789 | 0.339600 | 0.016846 | 0.102539 | 0.188407 |
| 19 | 0.033203 | 0.078369 | 0.064697 | 0.209229 | 0.010986 | 0.093506 | 0.189104 |
| 18 | 0.001953 | 0.031738 | 0.029053 | 0.151367 | 0.006592 | 0.067139 | 0.185547 |
| 17 | 0.012207 | 0.035889 | 0.027344 | 0.133789 | 0.000977 | 0.045410 | 0.156948 |
| 16 | 0.011963 | 0.046387 | 0.033203 | 0.151611 | 0.000000 | 0.018799 | 0.117327 |
| 15 | 0.001465 | 0.005615 | 0.004150 | 0.143066 | 0.000000 | 0.008057 | 0.098563 |
| 14 | 0.000000 | 0.010986 | 0.001709 | 0.036865 | 0.000244 | 0.002686 | 0.061942 |
| 13 | 0.000000 | 0.000488 | 0.000488 | 0.021240 | 0.000000 | 0.000488 | 0.027762 |
| 12 | 0.000000 | 0.009033 | 0.000000 | 0.005127 | 0.000000 | 0.000000 | 0.007673 |
| 11 | 0.000000 | 0.010498 | 0.000000 | 0.001709 | 0.000000 | 0.000000 | 0.004185 |
| 10 | 0.000000 | 0.007568 | 0.000000 | 0.000732 | 0.000000 | 0.000000 | 0.001953 |
| 9 | 0.001465 | 0.024902 | 0.000000 | 0.000244 | 0.000000 | 0.000000 | 0.000977 |
| 8 | 0.000000 | 0.000244 | 0.000000 | 0.000000 | 0.000000 | 0.000000 | 0.000767 |
| 7 | 0.000000 | 0.000000 | 0.000000 | 0.000732 | 0.000000 | 0.000000 | 0.000070 |
| 6 | 0.000000 | 0.000977 | 0.000000 | 0.000000 | 0.000000 | 0.000000 | 0.000209 |
| 5 | 0.000000 | 0.001221 | 0.000000 | 0.000244 | 0.000000 | 0.000000 | 0.000698 |
| 4 | 0.000000 | 0.000977 | 0.000000 | 0.000000 | 0.000000 | 0.000000 | 0.000488 |
| 3 | 0.000000 | 0.000000 | 0.000000 | 0.005615 | 0.000000 | 0.000000 | 0.000488 |
| 2 | 0.000000 | 0.003906 | 0.001709 | 0.000000 | 0.000000 | 0.000000 | 0.001256 |
| 1 | 0.022949 | 0.085449 | 0.004150 | 0.005859 | 0.000000 | 0.000000 | 0.000698 |
| 0 | 0.176270 | 0.303711 | 0.051514 | 0.023926 | 0.003418 | 0.004395 | 0.005232 |

Figure 5: Visualization of the magnitude of change during SFT for "math expert #2"

| | attention.wq.weight | attention.wk.weight | attention.wv.weight | attention.wo.weight | feed_forward.w1.weight | feed_forward.w3.weight | feed_forward.w2.weight |
|---|---|---|---|---|---|---|---|
| 31 | 0.774658 | 0.913086 | 0.894775 | 0.673828 | 0.495117 | 0.460693 | 0.291225 |
| 30 | 0.388428 | 0.662354 | 0.826904 | 0.700195 | 0.226074 | 0.183594 | 0.237374 |
| 29 | 0.127930 | 0.425293 | 0.483887 | 0.445312 | 0.086914 | 0.096436 | 0.207589 |
| 28 | 0.223145 | 0.379639 | 0.489014 | 0.612549 | 0.036621 | 0.056885 | 0.169992 |
| 27 | 0.049561 | 0.184814 | 0.572021 | 0.547363 | 0.010742 | 0.031738 | 0.133580 |
| 26 | 0.078369 | 0.168945 | 0.505371 | 0.582031 | 0.004395 | 0.014404 | 0.105050 |
| 25 | 0.014648 | 0.103027 | 0.439941 | 0.576416 | 0.003418 | 0.007080 | 0.070312 |
| 24 | 0.021729 | 0.143799 | 0.780029 | 0.663574 | 0.002930 | 0.004639 | 0.053920 |
| 23 | 0.018799 | 0.116943 | 0.715088 | 0.611328 | 0.002197 | 0.004883 | 0.056152 |
| 22 | 0.128662 | 0.242920 | 0.482422 | 0.603516 | 0.000977 | 0.004150 | 0.046108 |
| 21 | 0.223145 | 0.364990 | 0.437988 | 0.616455 | 0.001709 | 0.003662 | 0.044922 |
| 20 | 0.121826 | 0.156006 | 0.331299 | 0.487793 | 0.000488 | 0.002686 | 0.042969 |
| 19 | 0.051025 | 0.177979 | 0.405029 | 0.447510 | 0.000732 | 0.002686 | 0.053641 |
| 18 | 0.014404 | 0.043457 | 0.046143 | 0.181641 | 0.000732 | 0.001465 | 0.057408 |
| 17 | 0.017822 | 0.127686 | 0.082520 | 0.157715 | 0.000488 | 0.000732 | 0.054129 |
| 16 | 0.022217 | 0.060059 | 0.033691 | 0.202881 | 0.000000 | 0.000488 | 0.050432 |
| 15 | 0.029053 | 0.120117 | 0.016602 | 0.176270 | 0.000244 | 0.000732 | 0.041643 |
| 14 | 0.018311 | 0.006104 | 0.008301 | 0.040771 | 0.000244 | 0.000488 | 0.044015 |
| 13 | 0.001465 | 0.009277 | 0.009521 | 0.055664 | 0.000244 | 0.000488 | 0.037249 |
| 12 | 0.001953 | 0.017090 | 0.028809 | 0.038086 | 0.000000 | 0.000244 | 0.027623 |
| 11 | 0.001709 | 0.006348 | 0.015625 | 0.048096 | 0.000244 | 0.000244 | 0.027623 |
| 10 | 0.001221 | 0.007324 | 0.002441 | 0.023926 | 0.000000 | 0.000000 | 0.020299 |
| 9 | 0.000977 | 0.011475 | 0.004395 | 0.020996 | 0.000000 | 0.000000 | 0.010463 |
| 8 | 0.000732 | 0.003906 | 0.001953 | 0.016846 | 0.000000 | 0.000000 | 0.009696 |
| 7 | 0.000977 | 0.014648 | 0.002686 | 0.015137 | 0.000000 | 0.000000 | 0.007045 |
| 6 | 0.000977 | 0.009766 | 0.000732 | 0.010986 | 0.000000 | 0.000000 | 0.005232 |
| 5 | 0.002930 | 0.014648 | 0.000732 | 0.029541 | 0.000000 | 0.000000 | 0.005092 |
| 4 | 0.000488 | 0.006104 | 0.000732 | 0.027344 | 0.000000 | 0.000000 | 0.003697 |
| 3 | 0.000488 | 0.019531 | 0.000000 | 0.023438 | 0.000000 | 0.000000 | 0.002162 |
| 2 | 0.000488 | 0.003662 | 0.002197 | 0.003662 | 0.000000 | 0.000000 | 0.002232 |
| 1 | 0.074707 | 0.623779 | 0.148682 | 0.073242 | 0.000000 | 0.000000 | 0.002232 |
| 0 | 0.504150 | 0.694580 | 0.675537 | 0.324219 | 0.018311 | 0.021240 | 0.014718 |

Figure 6: Visualization of the magnitude of change during SFT for "Swahili expert #1"

| | attention.wq.weight | attention.wk.weight | attention.wv.weight | attention.wo.weight | feed_forward.w1.weight | feed_forward.w3.weight | feed_forward.w2.weight |
|---|---|---|---|---|---|---|---|
| 31 | 0.614258 | 0.919434 | 0.842773 | 0.633789 | 0.322998 | 0.332764 | 0.265206 |
| 30 | 0.260254 | 0.676270 | 0.936279 | 0.797852 | 0.123291 | 0.165283 | 0.157296 |
| 29 | 0.051025 | 0.489990 | 0.836914 | 0.591309 | 0.030029 | 0.079590 | 0.112863 |
| 28 | 0.024170 | 0.304688 | 0.785645 | 0.662109 | 0.003174 | 0.041748 | 0.086775 |
| 27 | 0.024658 | 0.356445 | 0.904053 | 0.727295 | 0.001709 | 0.026611 | 0.066895 |
| 26 | 0.015137 | 0.028809 | 0.831299 | 0.645264 | 0.000732 | 0.010986 | 0.045271 |
| 25 | 0.010986 | 0.265869 | 0.887939 | 0.645508 | 0.000732 | 0.002441 | 0.024972 |
| 24 | 0.006104 | 0.043701 | 0.941406 | 0.668213 | 0.000732 | 0.001465 | 0.015904 |
| 23 | 0.003906 | 0.042480 | 0.856201 | 0.519287 | 0.000732 | 0.001465 | 0.013811 |
| 22 | 0.005615 | 0.058350 | 0.670166 | 0.456787 | 0.000732 | 0.001709 | 0.015974 |
| 21 | 0.002930 | 0.014893 | 0.490479 | 0.389404 | 0.000732 | 0.001709 | 0.016183 |
| 20 | 0.029297 | 0.152832 | 0.734131 | 0.401367 | 0.000732 | 0.001221 | 0.014927 |
| 19 | 0.032959 | 0.094238 | 0.654785 | 0.385986 | 0.000732 | 0.000977 | 0.018485 |
| 18 | 0.000732 | 0.000732 | 0.097168 | 0.123291 | 0.000732 | 0.001221 | 0.025670 |
| 17 | 0.000977 | 0.003174 | 0.081787 | 0.149658 | 0.000732 | 0.001221 | 0.027762 |
| 16 | 0.000732 | 0.004639 | 0.042725 | 0.152588 | 0.000732 | 0.000732 | 0.024693 |
| 15 | 0.000732 | 0.000000 | 0.006104 | 0.106689 | 0.000732 | 0.000732 | 0.022740 |
| 14 | 0.000732 | 0.001709 | 0.002686 | 0.017334 | 0.000732 | 0.000732 | 0.021275 |
| 13 | 0.000732 | 0.000488 | 0.003906 | 0.051025 | 0.000732 | 0.000732 | 0.025112 |
| 12 | 0.000732 | 0.016846 | 0.026611 | 0.029053 | 0.000732 | 0.000732 | 0.022252 |
| 11 | 0.000732 | 0.000488 | 0.014893 | 0.043945 | 0.000732 | 0.000732 | 0.028948 |
| 10 | 0.000732 | 0.032471 | 0.006348 | 0.046387 | 0.000732 | 0.000732 | 0.024135 |
| 9 | 0.000732 | 0.015625 | 0.007324 | 0.042480 | 0.000732 | 0.000732 | 0.014439 |
| 8 | 0.000732 | 0.010498 | 0.001221 | 0.016846 | 0.000488 | 0.000244 | 0.011998 |
| 7 | 0.000732 | 0.011963 | 0.003906 | 0.030762 | 0.000244 | 0.000244 | 0.008580 |
| 6 | 0.000977 | 0.020996 | 0.000488 | 0.017090 | 0.000000 | 0.000244 | 0.006348 |
| 5 | 0.000244 | 0.006836 | 0.002686 | 0.037842 | 0.000000 | 0.000000 | 0.008789 |
| 4 | 0.000244 | 0.017822 | 0.000000 | 0.019287 | 0.000000 | 0.000000 | 0.005929 |
| 3 | 0.002930 | 0.075195 | 0.000000 | 0.024658 | 0.000000 | 0.000000 | 0.003418 |
| 2 | 0.001953 | 0.038818 | 0.014893 | 0.001953 | 0.000000 | 0.000000 | 0.002930 |
| 1 | 0.434082 | 0.920654 | 0.306396 | 0.121582 | 0.000000 | 0.000000 | 0.004534 |
| 0 | 0.395508 | 0.628174 | 0.808105 | 0.476074 | 0.018311 | 0.022705 | 0.021275 |

Figure 7: Visualization of the magnitude of change during SFT for "Japanese expert #1"

| | attention.wq.weight | attention.wk.weight | attention.wv.weight | attention.wo.weight | feed_forward.w1.weight | feed_forward.w3.weight | feed_forward.w2.weight |
|---|---|---|---|---|---|---|---|
| 31 | 0.580078 | 0.754150 | 0.893311 | 0.668701 | 0.283447 | 0.310791 | 0.362723 |
| 30 | 0.221924 | 0.152832 | 0.906006 | 0.625732 | 0.106201 | 0.161621 | 0.256557 |
| 29 | 0.024658 | 0.084717 | 0.670654 | 0.485596 | 0.019043 | 0.056885 | 0.165458 |
| 28 | 0.069580 | 0.050537 | 0.635986 | 0.640137 | 0.001709 | 0.012207 | 0.096191 |
| 27 | 0.040771 | 0.095459 | 0.927002 | 0.656494 | 0.000732 | 0.006104 | 0.056292 |
| 26 | 0.007324 | 0.015625 | 0.836670 | 0.624512 | 0.000488 | 0.004395 | 0.039551 |
| 25 | 0.022217 | 0.110840 | 0.853027 | 0.631836 | 0.000244 | 0.002197 | 0.025181 |
| 24 | 0.014893 | 0.037598 | 0.864746 | 0.667725 | 0.000488 | 0.002197 | 0.021275 |
| 23 | 0.007324 | 0.027100 | 0.852051 | 0.620361 | 0.000488 | 0.001709 | 0.015904 |
| 22 | 0.009766 | 0.006348 | 0.566406 | 0.507080 | 0.000488 | 0.001709 | 0.016462 |
| 21 | 0.024414 | 0.026367 | 0.607666 | 0.543213 | 0.000488 | 0.001953 | 0.017857 |
| 20 | 0.012451 | 0.014648 | 0.516357 | 0.492920 | 0.000244 | 0.000977 | 0.018555 |
| 19 | 0.008789 | 0.009277 | 0.819336 | 0.498047 | 0.000488 | 0.000732 | 0.019810 |
| 18 | 0.000732 | 0.000000 | 0.065918 | 0.194824 | 0.000244 | 0.000732 | 0.023368 |
| 17 | 0.000732 | 0.000000 | 0.158936 | 0.221436 | 0.000000 | 0.001465 | 0.029227 |
| 16 | 0.000732 | 0.002686 | 0.183350 | 0.248779 | 0.000732 | 0.000977 | 0.034110 |
| 15 | 0.000977 | 0.000488 | 0.177734 | 0.244629 | 0.001953 | 0.006104 | 0.050991 |
| 14 | 0.000977 | 0.000000 | 0.172852 | 0.194824 | 0.002197 | 0.007812 | 0.087193 |
| 13 | 0.000488 | 0.000000 | 0.319092 | 0.229248 | 0.002197 | 0.017334 | 0.112653 |
| 12 | 0.001953 | 0.000488 | 0.287842 | 0.156006 | 0.005615 | 0.040527 | 0.148926 |
| 11 | 0.000977 | 0.000000 | 0.395752 | 0.287842 | 0.003662 | 0.047363 | 0.161761 |
| 10 | 0.000488 | 0.000000 | 0.248535 | 0.212158 | 0.001953 | 0.022217 | 0.139230 |
| 9 | 0.001465 | 0.000000 | 0.258301 | 0.175293 | 0.000488 | 0.005859 | 0.098912 |
| 8 | 0.000977 | 0.000000 | 0.119629 | 0.130859 | 0.000977 | 0.008057 | 0.089774 |
| 7 | 0.003174 | 0.001709 | 0.144775 | 0.136719 | 0.000244 | 0.004639 | 0.066964 |
| 6 | 0.001709 | 0.000488 | 0.113770 | 0.128174 | 0.000244 | 0.005371 | 0.046875 |
| 5 | 0.000488 | 0.001221 | 0.166748 | 0.137695 | 0.000488 | 0.008057 | 0.041225 |
| 4 | 0.000732 | 0.002441 | 0.147217 | 0.193115 | 0.000732 | 0.007324 | 0.031599 |
| 3 | 0.000000 | 0.000244 | 0.005859 | 0.114502 | 0.000244 | 0.002930 | 0.014927 |
| 2 | 0.000244 | 0.010742 | 0.010742 | 0.090820 | 0.000244 | 0.000244 | 0.009696 |
| 1 | 0.012451 | 0.054199 | 0.036133 | 0.140381 | 0.000488 | 0.000488 | 0.009347 |
| 0 | 0.219482 | 0.442871 | 0.354248 | 0.345703 | 0.002197 | 0.006836 | 0.026925 |

Figure 8: Visualization of the magnitude of change during SFT for "Telugu expert #1"

| | attention.wq.weight | attention.wk.weight | attention.wv.weight | attention.wo.weight | feed_forward.w1.weight | feed_forward.w3.weight | feed_forward.w2.weight |
|---|---|---|---|---|---|---|---|
| 31 | 0.513428 | 0.614502 | 0.883789 | 0.732910 | 0.244385 | 0.261719 | 0.317104 |
| 30 | 0.149414 | 0.216797 | 0.880371 | 0.642822 | 0.075439 | 0.135254 | 0.226632 |
| 29 | 0.009033 | 0.107422 | 0.476318 | 0.439697 | 0.013184 | 0.060547 | 0.148996 |
| 28 | 0.019775 | 0.045898 | 0.637939 | 0.602051 | 0.001709 | 0.013916 | 0.098005 |
| 27 | 0.006836 | 0.035889 | 0.862793 | 0.667969 | 0.000977 | 0.011719 | 0.060965 |
| 26 | 0.004395 | 0.001953 | 0.765625 | 0.650391 | 0.000732 | 0.009033 | 0.038016 |
| 25 | 0.004150 | 0.011475 | 0.865723 | 0.648926 | 0.000488 | 0.002930 | 0.024623 |
| 24 | 0.006104 | 0.008789 | 0.905029 | 0.704346 | 0.000732 | 0.002686 | 0.021973 |
| 23 | 0.003906 | 0.008789 | 0.906738 | 0.680176 | 0.000732 | 0.001953 | 0.018834 |
| 22 | 0.000977 | 0.007568 | 0.617676 | 0.532715 | 0.000732 | 0.001465 | 0.020229 |
| 21 | 0.008057 | 0.003174 | 0.593506 | 0.529297 | 0.000488 | 0.002197 | 0.020438 |
| 20 | 0.002930 | 0.015137 | 0.479248 | 0.525879 | 0.000488 | 0.001465 | 0.024484 |
| 19 | 0.002197 | 0.006104 | 0.707764 | 0.530273 | 0.000244 | 0.000732 | 0.026018 |
| 18 | 0.000732 | 0.000000 | 0.168213 | 0.235596 | 0.000244 | 0.001221 | 0.034598 |
| 17 | 0.000732 | 0.000488 | 0.165527 | 0.247070 | 0.000244 | 0.001709 | 0.044364 |
| 16 | 0.000977 | 0.000244 | 0.249512 | 0.305176 | 0.000244 | 0.001953 | 0.057687 |
| 15 | 0.002197 | 0.003662 | 0.304688 | 0.288574 | 0.000488 | 0.010010 | 0.081124 |
| 14 | 0.005127 | 0.000244 | 0.323242 | 0.381348 | 0.002441 | 0.028809 | 0.137695 |
| 13 | 0.001221 | 0.000000 | 0.443115 | 0.348389 | 0.003662 | 0.059082 | 0.190081 |
| 12 | 0.000977 | 0.005859 | 0.502930 | 0.293457 | 0.005615 | 0.064697 | 0.203544 |
| 11 | 0.001221 | 0.000244 | 0.515137 | 0.375732 | 0.011230 | 0.112549 | 0.235421 |
| 10 | 0.000488 | 0.000000 | 0.340576 | 0.318848 | 0.003662 | 0.061279 | 0.187849 |
| 9 | 0.001709 | 0.000244 | 0.281006 | 0.264648 | 0.000977 | 0.028076 | 0.129325 |
| 8 | 0.002441 | 0.000488 | 0.162354 | 0.208740 | 0.000244 | 0.018799 | 0.092564 |
| 7 | 0.002686 | 0.000977 | 0.145020 | 0.174072 | 0.000000 | 0.012451 | 0.072266 |
| 6 | 0.000977 | 0.000000 | 0.091309 | 0.163574 | 0.000000 | 0.005859 | 0.042201 |
| 5 | 0.000244 | 0.000000 | 0.075684 | 0.109863 | 0.000244 | 0.003662 | 0.030901 |
| 4 | 0.000244 | 0.002197 | 0.015869 | 0.126221 | 0.000000 | 0.001953 | 0.022670 |
| 3 | 0.000244 | 0.001221 | 0.001465 | 0.093262 | 0.000000 | 0.000732 | 0.008440 |
| 2 | 0.001465 | 0.009277 | 0.005371 | 0.039551 | 0.000244 | 0.000244 | 0.005232 |
| 1 | 0.074463 | 0.338135 | 0.019287 | 0.082031 | 0.000000 | 0.000244 | 0.005720 |
| 0 | 0.258301 | 0.520508 | 0.572998 | 0.323242 | 0.007568 | 0.009521 | 0.026158 |

Figure 9: Visualization of the magnitude of change during SFT for "Bengali expert #1"

