# OpenReview forum: "Layer Swapping for Zero-Shot Cross-Lingual Transfer in Large Language Models"
_ICLR.cc/2025/Conference — ICLR 2025 Spotlight_

### Official Review · Reviewer_X6PS · 2024-11-01

**Soundness:** 3
**Presentation:** 3
**Contribution:** 3
**Rating:** 8
**Confidence:** 4

**Summary:**

The paper introduces a novel layer-swapping methodology for zero-shot cross-lingual transfer in large language models (LLMs) aimed at improving task performance in non-English languages, particularly for mathematical reasoning. The approach tackles the lack of task-specific data in low-resource languages by fine-tuning two separate "experts" of the same base model: one trained on English mathematical data and another trained on general instruction data in the target language. The proposed method then selectively replaces the top and bottom transformer layers of the math expert with those from the language expert, buffered by transition zones between these regions. This configuration shows promising performance gains on the MGSM math benchmark across four languages—Swahili, Telugu, Bengali, and Japanese—without any additional in-language math data.

**Strengths:**

- The paper introduces an efficient, innovative layer-swapping method for zero-shot cross-lingual transfer in LLMs, addressing the lack of task-specific data in low-resource languages with simplicity and strong empirical results.

- This technique is particularly notable for its straightforward implementation, allowing effective merging of task and language expertise without complex adjustments, making it a practical alternative to standard methods like model souping.

- Promising experimental gains on math reasoning benchmarks across multiple low resource languages validate the method's effectiveness, showing that layer swapping successfully enhances cross-lingual transfer without in-language task data.

**Weaknesses:**

- The method is tested only on math reasoning, leaving it unclear if layer swapping generalizes to other tasks. Additional evaluations on tasks like question-answering or translation would strengthen the claims of broad applicability.
- While the paper mentions different layer-swapping configurations, it lacks in-depth analysis on which configurations work best and why. A more detailed study of these choices would help to better understand the method make it more robust. For example, provide ablation studies on the number of swapped layers or transition zone sizes, or to analyze how performance changes as these parameters are varied.
- Comparisons to recent modular fine-tuning techniques, such as adapters or LoRA, are missing. Including these would clarify how layer swapping performs relative to other efficient, cross-lingual methods.

**Questions:**

1. Can the authors clarify if they have tried layer swapping on other tasks, such as translation, question-answering, or code generation? Evidence of generalizability beyond math would make the approach significantly stronger.
2. Layer swapping is positioned as an alternative to methods like model souping, but a comparison to more recent modular fine-tuning techniques, such as adapters (Pfeiffer et al., 2020) or LoRA (Hu et al., 2022), could help contextualize its relative strengths. Can the authors either conduct these comparisons (e.g. computational efficiency, performance) or discuss the anticipated performance differences?
3. Given that this study uses a 8B (32-layer) LLM, how would the authors anticipate the method scaling to models with more layers or parameters? Could they provide guidance on applying layer swapping to larger models, especially in terms of choosing the number of layers to swap? Would the authors still anticipate the same performance gain with layer swapping on larger models?
4. For Japanese, layer swapping results in lower average performance compared to the individual math experts. The authors mentioned that the Japanese experts were the weakest as performance across BELEBELE, FLORES, MBPP, and MMLU were minimal. Could the authors share the results on these benchmark (before and after SFT) to help better understand the case?

---

> ### Author Response · Authors · 2024-11-23
> **Response to Reviewer X6PS**
>
> We would like to thank the reviewer for their affirmation that this is a useful contribution to the field as well as their attentive feedback and suggestions. We address their concerns below, but the TLDR is that we were limited by the availability of difficult task-specific benchmarks in medium-/low-resource languages and that our configuration ablation studies were not discussed due to lack of statistical significance.
>
> Weaknesses:
> - While only evaluating on math reasoning does limit the generalizability of the findings, we chose math (MGSM) because we wanted a somewhat difficult benchmark (at least more difficult than MBPP, XLSum, Belebele, FLORES, etc.) which required reasoning and task/domain-specific competence. There are very few such benchmarks that cover more than a few languages, and we did not want to rely on machine-translated benchmarks. On other benchmarks we looked at, the gap in performance between English and other languages was much lower. We will mention the limited multilingual benchmarks available in the limitations section and discuss that we looked at m-MMLU and MBPP.
> - Our limited discussion of our ablation studies into the possible layer-swapping configurations is in large part because of the statistical insignificance of these tests. We performed ablation studies across many dimensions but as mentioned in the paper, in a very large range, the results were more or less the same (in other words, equally effective). We chose to therefore, only discuss some trends we saw without devoting too much space in the paper on it. On the flip side, we argue that the large range of configurations in which this worked to be a strength → that this method is not overly sensitive to configurations.
> - While such modular solutions like such as MAD-X adapters or LoRA are quite relevant, we argue they are used in very different settings. We note that initially the goal of this project was to explore model merging and resolving interference specifically in the context of cross-lingual task transfer. It was in large part, inspired by the popularity of model souping, and the many benefits post-hoc merging could bring. As a result, the experimental setup was from the beginning about merging “experts” *without retraining*. Comparisons to MAD-X, LoRA, or other solutions would require re-formulating the problem at hand, as these require knowing beforehand exactly how to set up model finetuning and are not nearly as flexible as this proposed solution. However, we do believe the success of this method has potential impacts on designing modular finetuning methods, such as freezing the middle layers during cross-lingual transfer. We can clarify this in the paper to better justify the positioning of this methodology in broader literature.
>
> Questions:
> 1. We address this in the first bullet point above, but the short answer is no — MGSM was the target task for this project.
> 1. Similarly we address this in the last bullet point above, but we argue that posthoc model merging is addressing a different finetuning scenario than other such methods.
> 1. Our intuition is that this method would scale up to larger model sizes that have similar English-centric pretrainings. This because we believe this method works because such models “think in English” (process tokens in English-centric representation spaces) and relevant literature finds this to be the case for models of different sizes (https://aclanthology.org/2024.acl-long.820.pdf, https://aclanthology.org/2024.acl-long.309.pdf, https://aclanthology.org/2024.naacl-long.384.pdf, https://openreview.net/forum?id=fR3wGCk-IXp). For larger models that are more capable, this method may be less effective for the same languages that presumably would have higher math performance to begin with in the larger model. However, this method would perhaps therefore be more applicable to lower-resource languages. We, however, were unable to run such experiments because of resource constraints and therefore choose to not generalize the conclusions of our experiments to such models.
> 1. Yes, we will provide a full table in the appendix. To summarize, The language experts for Swahili, Bengali, and Telugu increased their Belebele by 2-4 points & Flores scores by 1 BLEU typically for those languages, also with slight increases on MBPP & MMLU. In comparison, the Japanese experts didn’t see any increase by more than 1 point (no regression, but at most small increase) for the different tasks.

---

> > ### Comment · Reviewer_X6PS · 2024-11-25
> >
> > Thanks the authors for the response. Most of my concerns have been addressed, and I decide to raise my overall rating from 6 --> 8 after reading your response and the updated paper. Your method also remind me of a very recent work on uncovering where multi-task learning happens in LLMs [1], the paper pointed out that instruction-tuned LLMs organize their layers into three functional groups: early shared layers for general features, middle transition layers for task-specific transitions, and final layers for refinement. The findings complement the idea of shared layers and transition layers in LLMs, although they only looked at English setting tasks.
> >
> > [1] Layer by Layer: Uncovering Where Multi-Task Learning Happens in Instruction-Tuned Large Language Models https://aclanthology.org/2024.emnlp-main.847.pdf

---

### Official Review · Reviewer_oaJT · 2024-11-04

**Soundness:** 4
**Presentation:** 3
**Contribution:** 2
**Rating:** 6
**Confidence:** 4

**Summary:**

The paper introduces a variant of model merging addressing the challenge of adapting LLMs for tasks in low-resource languages. The methodology involves *layer swapping* (parameters being swapped) between a task expert and a language expert, both following the same underlying architecture. The resulting re-composed LLM is said to outperform both the individual experts without the requirement of a task-specific fine-tuning in the low-resource language. The experiments involve evaluation of the proposed methodology on MGSM (a multilingual benchmark of grade-school math problems) on 4 resource-scarce languages: Japanese, Telugu, Swahili and Bengali.

**Strengths:**

1. The proposed methodology is highly practical in scenarios where one might have publicly available task-specific data in a high-resource language and generic instruction data in the low-resource language. The model parameter adjustments being fully post-hoc eliminate any additional computational overhead apart from the initial fine-tuning required to create task and language experts.

2. *layer swapping* with the best configuration consistently outperforms the individual SFT experts, the base LLM Llama 3.1 and the general model souping approach in three (Swahili, Telugu, Bengali) out of the four languages under this study.

3. The paper is easy to follow. The authors also take the effort to acknowledge the possible limitations to the work, encouraging future exploration.

**Weaknesses:**

1. The methodology is evaluated only on Llama 3.1, using MGSM benchmark for 4 selective languages. In my opinion, evaluation of the method on a single model, single benchmark and limited languages makes the conclusion less generalizable. While the languages used in the study are diverse, incorporating more datasets and models (in terms of different architecture or pre-training) can strengthen the conclusion.

2. The assumption of availability of generic instruction data for low-resource languages might not hold for all languages. Task-specific data and generic instruction data in a high-resource language is generally more accessible. An experiment where language expert is fine-tuned using translated instructions would increase the practicality of the work.

**Questions:**

1. How does a re-composed model in one language affect model performance in typologically similar languages? In my opinion, an analysis of this kind would highly benefit the work.

2. Would a 2-stage SFT work better than Joint SFT on language and task?

---

**[POST-REBUTTAL UPDATE]**

I have decided to increase the overall rating from **5** (marginally below the acceptance threshold) to **6** (marginally above the acceptance threshold).

---

> ### Author Response · Authors · 2024-11-23
> **Rebuttal to Reviewer oaJT**
>
> We thank you for your thoughtful review, with constructive comments and positive feedback. We provide responses to your enumerated comments below. The TLDR is that we argue that our paper presents a substantial contribution despite what the reviewer describes as insufficient experiment, pointing to its grounding in other interpretability papers. We also note the limited available evaluation tasks in medium-resource languages.
>
> Weaknesses:
> 1. We acknowledge that our conclusions may be considered limited in its generalizability because of the experiments we were able to conduct. However we want to respond with several points:
>
> a. We chose math (MGSM) because we wanted a somewhat difficult benchmark (at least more difficult than MBPP, XLSum, Belebele, FLORES, etc.) which required reasoning and task/domain-specific competence. There are very few such benchmarks that cover more than a few languages, and we did not want to rely on machine-translated benchmarks. On other benchmarks we looked at, the gap in performance between English and other languages was much lower. We will mention the limited multilingual benchmarks available in the limitations section and discuss that we looked at m-MMLU and MBPP.
>
> b. We argue that generalizability of this result is probable because of how our findings align with recent literature in multilingual interpretability. Through different means sever papers conclude that the the top & bottom layers are the  most significant to the multilingual capabilities of English-centric models (https://aclanthology.org/2024.acl-long.820.pdf, https://aclanthology.org/2024.acl-long.379/, https://openreview.net/forum?id=fR3wGCk-IXp). Given feedback from other reviewers, we will strengthen our grounding of these findings in academic literature to justify the significance of the results.
>
> c. In the paper, we are careful to not claim beyond what we have tested and intentionally use language such as “conjecture” and “hypothesize”. In addition, we discuss further experimentation in Limitations & Future Work.
>
> d. As displayed in the Limitations & Future Work section, the success of this method in this situation raises many questions. We view the large number of experiments that can be derived from these findings to be indicative of the significance of this method and its contribution to the research community.
>
> 2. It is a fair point that not in all languages can we find instruction data, and the lower-resource the language, the harder this will be to find. In the mixes of instruction data we assembled however, machine translation samples were the most common. Machine translation data is in abundance for even lower-resource languages, but it remains to be seen how effective this method would work with *only* finetuning on MT data. Nonetheless, finding instruction data for medium- and low-resource languages of any sort is still easier than task-specific data, and we anticipate this to be the case as more resources are created in more languages.
>
> Questions:
> 1. That does seem like an interesting analysis. The only results we have are that for the Japanese layer-swapped model, math performance in Chinese was about the same as the math expert on its own. Similarly for the Swahili layer-swapped model, math results in French and Spanish are also equivalent to the math expert. These language combinations are very distantly related of course, so this doesn’t really answer the question. We note that the MGSM benchmark covers only about 10 languages and purposefully covers a diverse set of languages from different families, and therefore this sort of analysis may require the creation of a whole new evaluation dataset.
> 1. It’s certainly possible. In doing a hyperparameter search for the joint SFT baseline, we had initially had this setup, but moved to mixed data which ended up being better. Unclear if this is caused by the data or the hyperparameter setup.

---

> > ### Comment · Reviewer_oaJT · 2024-11-25
> >
> > I thank the authors for their responses to my comments and questions. They have addressed most of my questions. I appreciate them for their effort to update the paper draft, incorporating all the review feedbacks. With this thought, I have decided to increase the overall rating of the paper. Please refer to my review for the revised scores.

---

### Official Review · Reviewer_edMm · 2024-11-07

**Soundness:** 4
**Presentation:** 4
**Contribution:** 3
**Rating:** 8
**Confidence:** 4

**Summary:**

This paper proposes a new approach to perform zero-shot cross-lingual transfer for solving tasks in a new language. The requirement is to have data for the task in a more resourced language (e.g. English) and non-task-specific data for other languages (4 languages in this paper). The idea is to fine-tune the model (Llama 3.1, 8B in the paper) separately to one new language and another copy to the task in English, then compose a new model of layers of these two models. A study is included of how much the models change during fine-tuning, which concludes that math tasks cause changes in the fine-tuned model closer to the middle layers, while tuning on a new language causes changes in the first and last layers. Based on that the composed model takes task-layers from the math-tuned model's middle, and language-layers from the language-tuned model's start and end. Transfer between the layers of two independently tuned models is done by "souping" intermediate layers, i.e. averaging the weights of layers of both models. Gains up to 10% are shown, in comparison to the expert models.

**Strengths:**

* Interesting idea and its evaluation on 1 model and 4 languages, with additional experiments
* Although the setup raises some questions (limited evaluation, why not freeze layers and avoid having to soup the transition layers, etc.), the expanded evaluation on Swahili and the limitations section address most of these
* excellent writing, justification and presentation

**Weaknesses:**

* limited evaluation: only 1 model and one set of tasks (math)

**Questions:**

1. You write that combining layers works because models are fine-tuned for just a little bit: why not tune for longer?
2. What would the intuitive explanation be behind the findings of your preliminary analysis on which layers are changed by task-tuning and language-tuning, other examples of similar effects in related literature?
3. If you were to tune for longer, would that affect the results of the preliminary analysis and change more layers, or different layers?

---

> ### Author Response · Authors · 2024-11-23
> **Response to Reviewer edMm**
>
> We would like to thank the reviewer for their mix of constructive feedback and affirmation that this is a useful contribution to the field.
> Strengths:
> Thanks for the positive feedback. To give extra context about the “why not freeze layers”, the goal of this project was to explore model merging and resolving interference in the context of cross-lingual task transfer. It was in large part, inspired by the popularity of model souping, and the many benefits post-hoc merging could bring. As a result, the experimental setup was from the beginning about merging “experts” without retraining. The downside of freezing beforehand would be needing to know beforehand what configuration of layers would work. Therefore you would likely have to finetune many times to find the optimal setup versus here we can cheaply iterate through different merging methods (similar to why model souping is practical). However, we leave freezing layers before finetuning to future work, because the success of this method certainly implies potential success there as well.
>
> Weaknesses:
> We acknowledge that our evaluations can be considered limited given the success of this method. We wanted to bring up a few points, however:
> 1. We chose math (MGSM) because we wanted a somewhat difficult benchmark (at least more difficult than MBPP, XLSum, Belebele, FLORES, etc.) which required reasoning and task/domain-specific competence. There are very few such benchmarks that cover more than a few languages, and we did not want to rely on machine-translated benchmarks. On other benchmarks we looked at, the gap in performance between English and other languages was much lower. We will mention the limited multilingual benchmarks available in the limitations section and discuss that we looked at m-MMLU and MBPP.
> 1. In the paper, we are careful to not claim beyond what we have tested and intentionally use language such as “conjecture” and “hypothesize”. In addition, we discuss further experimentation in Limitations & Future Work.
> 1. As displayed in the Limitations & Future Work section, the success of this method in this situation raises many questions. We view the large number of questions that are prompted from these findings to be indicative of the significance of this method and its contribution to the research community.
>
> Questions:
> 1. We acknowledge that the success of this method raises a large number of followup questions about the extent of the settings in which this would work and doing longer SFT runs is certainly an ablation we are curious about. However, we also note that we were limited by the availability of high quality, open-source instruction tuning data in each of the target languages, especially ensuring a mix of tasks. We also saw with higher learning rates (during our hyperparameter search), that SFT was not quite as effective. In the paper we noted that training on 30-40k samples is not large, relative to some modern SFT regimes. But in multilingual settings, it is often the case that you do not have more data available than this and so it is perhaps not small in that sense.
> 1. Our intuition is that the “internal workings” of Llama 3 (and other similarly English-centric LLMs) is itself in English, and so it might take a couple of the edge layers to map input/output tokens in a non-English language to these English-centric representations. A number of recent interpretability papers find this directly either by tracking representations (https://aclanthology.org/2024.acl-long.820.pdf, https://arxiv.org/abs/2410.09223) or identifying language-specific parameters (https://aclanthology.org/2024.acl-long.309.pdf, https://aclanthology.org/2024.naacl-long.384.pdf), most of which are in the top and bottom layers. In addition, a number of recent papers find success in encouraging the model to “think in English” when doing multilingual tasks, either during finetuning or at inference (https://aclanthology.org/2024.acl-long.379/, https://openreview.net/forum?id=fR3wGCk-IXp). It is good feedback that we may want to better ground our findings in academic literature, especially given the breadth of very recent papers on the subject. We will work to do so.
> 1. It certainly is possible that finetuning for longer (more data, more epochs, or stronger learning rate) could lead to different results. However, we note that the visualizations and analysis leveraged relative parameter change magnitudes. Longer finetuning could potentially reduce noise in the process, but we additionally reiterate the strong consistency of the findings of this analysis across many training runs and many languages.

---

### Author Response · Authors · 2024-11-24
**Paper Edits for Rebuttal Revision**

We have updated the paper PDF to present our Rebuttal Revision, which updates the original paper by taking into account the thoughtful feedback of the 3 reviewers.

Our edits are summarized as:
1. We ground the method in academic literature across several sections, providing more intuition for why it works. The TLDR is that recent work shows English-centric LLMs "think in English" and for multilingual text, the most important layers are the top & bottom, which map multilingual input/output to and from English representation spaces. This addresses all 3 reviewers' feedback.
1. We provide results of the individual experts on our benchmarks, as requested by reviewer X6PS.
1. We add to our limitations more context for why MGSM was our target task in this paper. This addresses all 3 reviewers' feedback.
1. We cut a few paragraphs in order to still fulfill 10-page maximum.

We hope that with these updates will prompt the reviewers to update their reviews, as well.

---

### Meta-Review · Area_Chair_Rwc2 · 2024-12-19

**Metareview:**

The paper introduces a model merging methodology tailored to address the challenge of adapting large language models (LLMs) to perform tasks in non-English languages without task-specific data. The method involves independently fine-tuning "experts" on math tasks in English and generic instruction data in target languages, followed by swapping top and bottom layers from the language expert into the math expert. This approach achieves up to a 10% performance gain on the MGSM benchmark for mathematical reasoning in resource-scarce languages such as Swahili, Bengali, Telugu, and Japanese. Reviewers largely commended the novelty and practicality of this approach, emphasizing its simplicity, interpretability, and effectiveness in enabling post-hoc modularity in multilingual tasks. Strengths include the innovative model recomposition mechanism, compelling empirical results, and clear presentation. Weaknesses noted by reviewers include the limited evaluation scope—restricted to one benchmark, four languages, and one base model (LLaMA 3.1)—as well as reliance on instruction-tuning data for target languages and a lack of detailed ablation studies or comparisons with alternative modular methods like LoRA. The acceptance decision is grounded in the paper's methodological originality, empirical contributions, and potential impact in advancing cross-lingual capabilities for low-resource languages, particularly in settings with limited task-specific data availability.

**Additional Comments On Reviewer Discussion:**

During the rebuttal period, reviewers raised a number of concerns and provided feedback on evaluation scope, methodological comparisons, and generalizability. Reviewer edMm noted limited evaluation on a single model (LLaMA 3.1), one benchmark (MGSM), and four languages. The authors explained constraints in multilingual benchmarks and pointed to alignment with existing literature, emphasizing that their findings were carefully framed as hypotheses and providing detailed explanations for their choices. Reviewer oaJT also suggested incorporating more models and datasets, particularly exploring typological language effects and the practicality of instruction data. The authors acknowledged these limitations but highlighted the scarcity of comparable multilingual datasets and shared additional insights, such as typological transfer observations in related language pairs. Reviewer X6PS noted missing comparisons to modular fine-tuning methods (e.g., LoRA, MAD-X) and ablation studies on swapping configurations. The authors clarified that their post-hoc merging approach differs fundamentally from these methods and argued that the lack of sensitivity to configurations strengthens the method’s applicability.
The reviewers generally agreed that the authors addressed most concerns effectively, with two raising their scores post-rebuttal. The explanations on benchmark selection, literature grounding, and flexibility of the method were particularly persuasive. However, the lack of broader task evaluations remained a limitation. In the final decision, the rebuttals and revisions were weighed against the paper's originality, simplicity, and demonstrated gains. The explanations provided sufficient confidence in the method's potential impact despite the scope limitations. The decision to recommend acceptance reflects the consensus on the paper's contributions to modular LLM design for cross-lingual tasks and its alignment with emerging research trends.

---

### Decision · Program_Chairs · 2025-01-22

Accept (Spotlight)